# Thawing Yedoma permafrost is a neglected nitrous oxide source

M. E. Marushchak [1,13 ✉], J. Kerttula[1], K. Diáková [1,2], A. Faguet[3], J. Gil[1,4], G. Grosse [5,6], C. Knoblauch [7,8], N. Lashchinskiy[3,9], P. J. Martikainen [1], A. Morgenstern [5], M. Nykamb[1], J. G. Ronkainen[1], H. M. P. Siljanen [1,10], L. van Delden [1,5], C. Voigt [1,11], N. Zimov[12], S. Zimov [12] & C. Biasi [1]

In contrast to the well-recognized permafrost carbon (C) feedback to climate change, the fate of permafrost nitrogen (N) after thaw is poorly understood. According to mounting evidence, part of the N liberated from permafrost may be released to the atmosphere as the strong greenhouse gas (GHG) nitrous oxide ($N_2O$). Here, we report post-thaw $N_2O$ release from late Pleistocene permafrost deposits called Yedoma, which store a substantial part of permafrost C and N and are highly vulnerable to thaw. While freshly thawed, unvegetated Yedoma in disturbed areas emit little $N_2O$, emissions increase within few years after stabilization, drying and revegetation with grasses to high rates (548 (133–6286) µg N m$^{-2}$ day$^{-1}$; median with (range)), exceeding by 1–2 orders of magnitude the typical rates from permafrost-affected soils. Using targeted metagenomics of key N cycling genes, we link the increase in in situ $N_2O$ emissions with structural changes of the microbial community responsible for N cycling. Our results highlight the importance of extra N availability from thawing Yedoma permafrost, causing a positive climate feedback from the Arctic in the form of $N_2O$ emissions.

[1] Department of Environmental and Biological Sciences, University of Eastern Finland, Kuopio, Finland. [2] Department of Soil Biogeochemistry, Bayreuth Center of Ecology and Environmental Research (BayCEER), University of Bayreuth, Bayreuth, Germany. [3] Trofimuk Institute of Petroleum Geology and Geophysics, Novosibirsk, Russia. [4] Department of Integrative Biology, Great Lakes Bioenergy Research Center, Michigan State University, East Lansing, MI, USA. [5] Alfred Wegener Institute Helmholtz Centre for Polar and Marine Research, Potsdam, Germany. [6] Institute of Geosciences, University of Potsdam, Potsdam, Germany. [7] Institute of Soil Science, Universität Hamburg, Hamburg, Germany. [8] Center for Earth System Research and Sustainability, Universität Hamburg, Hamburg, Germany. [9] Central Siberian Botanical Garden, Novosibirsk, Russia. [10] Department of Functional and Evolutionary Ecology, University of Vienna, Vienna, Austria. [11] Department of Geography, University of Montreal, Montreal, QC, Canada. [12] North-East Scientific Station, Pacific Institute for Geography, Far-East Branch, Russian Academy of Sciences, Cherskii, Russia. [13] Present address: Department of Biological and Environmental Science, University of Jyväskylä, Jyväskylä, Finland. ✉email: maija.marushchak@uef.fi

Rapid Arctic warming[1] and associated permafrost thaw[2,3] are threatening the large C and N reservoirs of northern permafrost soils[4–6], accumulated under cold conditions where the decomposition rate of soil organic matter (SOM) is low[7,8]. Permafrost thaw is now increasingly exposing these long-term inert C and N pools to microbial decomposition and transformation processes. While it is long known that mobilization of permafrost C potentially increase the release of the greenhouse gases (GHG) carbon dioxide ($CO_2$) and methane ($CH_4$)[5,9,10], the fate of soil N liberated upon permafrost thaw is poorly studied and more complex. There is evidence that part of liberated N may be emitted to the atmosphere as nitrogenous gases, most importantly as $N_2O$[6], which is a ~300 times more powerful GHG than $CO_2$ over a 100-year time horizon[11] and a dominant contributor to ozone destruction in the stratosphere[12].

The current increase in atmospheric $N_2O$ concentration is mainly driven by the growth of human-induced emissions, which comprise 43% of the global $N_2O$ emissions of 17.0 Tg N year$^{-1}$ and are dominated by $N_2O$ release from fertilized agricultural soils[13]. Nitrous oxide emissions, although generally smaller per unit area, occur also from soils under natural vegetation with a 33% contribution to the total global $N_2O$ emission[13]. Tropical soils with high N turnover rates generally show the largest $N_2O$ emissions among natural soils, while permafrost-affected soils in cold environments have been thought to be negligible $N_2O$ sources. This view was challenged by a recent synthesis showing that $N_2O$ emissions commonly occur from permafrost soils, with a global emission between 0.08 and 1.27 Tg N year$^{-1}$, meaning a 1–23% addition to the global $N_2O$ emission from natural soils[6]. However, this estimate is still highly uncertain due to the overall scarcity of $N_2O$ flux observations from permafrost-affected soils and the lack of studies from some important permafrost soil types, including the Yedoma studied here.

Late-Pleistocene aged Yedoma permafrost occurs as deep deposits (a mean thickness of ~19 m) over an area of > 1 million km$^2$ in the Northern Hemisphere (Fig. 1)[14]. The Yedoma region contains >25% of the circumarctic permafrost C stock[15], and a yet unaccounted for and likely even larger proportion of permafrost N because of the low C/N ratio of Yedoma SOM (typically < 15)[16,17]. The SOM in Yedoma is thought to be easily decomposable because it was incorporated into the permafrost soon after deposition without having much time to be degraded[15]. The high ice content of Yedoma[14] makes it vulnerable for abrupt thaw and ground collapse[18], allowing rapid mobilization of soil C and N stocks after thaw[15]. Along Arctic rivers and the coastal zone of the Arctic Shelf, thawing of Yedoma permafrost creates steep, tens-of-meters-high Yedoma exposures[19–21], where many of the conditions known to promote $N_2O$ emissions from permafrost-affected soils[6] are met, including low C/N ratios, lack of vegetation, and suitable soil moisture content for microbial processes producing $N_2O$.

Here, we studied $N_2O$ fluxes on two thawing Yedoma exposures forming retrogressive thaw slumps in Northeast Siberia: In July 2016 on Kurungnakh Island situated in the Lena River Delta and in July 2017 in Duvanny Yar located by the Kolyma River (See Methods section; Fig. 1). At both sites, we measured $N_2O$ fluxes with the static chamber technique[22] and determined the mineral N pools on transects spanning from the top of the thawing Yedoma exposure across the bare and revegetated parts down to the river shore (see Methods section; Supplementary Figs. 1 & 2). At the intensive study site Kurungnakh we additionally studied N transformation and $N_2O$ production rates in the laboratory, as well as the relative abundance of N cycling genes. We revealed an increasing trend in $N_2O$ emissions with drying, stabilization, and revegetation of the thawed Yedoma sediments. Increased emissions were coupled with changes in the microbial community composition responsible for soil N transformation processes.

## Results and discussion

**Nitrous oxide emissions and mineral N pools across thawing Yedoma exposures.** Our field flux measurements revealed substantial $N_2O$ release from Yedoma permafrost following thaw. At the Kurungnakh exposure, the $N_2O$ fluxes from thawed Yedoma surfaces were highly variable (63 (–19–6286) µg N m$^{-2}$ day$^{-1}$; median with (range)), at the high-end exceeding the typical fluxes from permafrost-affected soils (38 (6–189) µg N m$^{-2}$ day$^{-1}$; median with (25th–75th percentiles); ref. [6]) by two orders of magnitude. The $N_2O$ emissions showed an increasing trend along the measuring transect (Fig. 2a, Supplementary Table 1) from the densely vegetated Holocene cover deposits overlying intact permafrost on the top of the riverbank, through the actively eroding upper part of the Yedoma exposure, down to the already stabilized lower part of the slope revegetated by mosses and grasses. The highest $N_2O$ emissions occurred from Yedoma which had thawed between 5 and 10 years ago (see Methods; Supplementary Fig. 1e, Supplementary Fig. 3) and that were revegetated by grasses (548 (133–6286) µg N m$^{-2}$ day$^{-1}$; mean with (range)). The emissions from these revegetated Yedoma soils in the mid part of the slope were significantly higher than the emissions from undisturbed vegetated Holocene cover and bare freshly thawed Yedoma (Dunn's test, $p < 0.05$). Negligible $N_2O$ fluxes were detected from bare sand on the river shore that receives melt waters from the thawing Yedoma exposure above (Fig. 2a, Supplementary Table 1).

The spatial pattern in the $N_2O$ emissions in situ was confirmed by laboratory incubations with Kurungnakh soils, where the highest $N_2O$ production under anoxic conditions was found in Yedoma revegetated with grasses and the lowest in bare earlier thawed Yedoma and vegetated Holocene cover (Kruskall–Wallis test, $p = 0.0004–0.007$; Fig. 3a, Supplementary Table 4). Higher $N_2O$ production under the anoxic treatment than in the oxic treatment (Wilcoxon signed-rank test, $p < 0.0001$) shows that denitrification is the main $N_2O$ production pathway. The yet higher $N_2O$ production in anoxic conditions in the presence of acetylene in all soils except bare freshly thawed Yedoma (1.7 to 4.7 times depending on the surface type; Wilcoxon signed-rank test, $p = 0.01$) suggests that, in addition to $N_2O$, a significant amount of $N_2$ is emitted from the studied soils to the atmosphere.

The $N_2O$ fluxes at the Duvanny Yar exposure were less variable and lower than at the Kurungnakh exposure (Kruskall–Wallis test, $p = 0.047$), ranging from –147 to 222 µg N m$^{-2}$ day$^{-1}$ (Fig. 2a, Supplementary Table 1). The highest $N_2O$ emissions were again detected from thawed and revegetated Yedoma surfaces, but this time from those with mosses. There, the $N_2O$ emissions were significantly higher than from bare freshly thawed Yedoma (Dunn's test, $p = 0.010$). Further, we found elevated $N_2O$ concentrations (mean ± SD 2.4 ± 2.1 ppm; compared to atmospheric $N_2O$ concentration of 0.33 ppm) in the soil pore gas in Yedoma covered by high dead standing biomass of the pioneering plant *Descurainia sophioides* (Supplementary Fig. 5a). We estimated (Supplementary Fig. 5b) the $N_2O$ flux at these surfaces at 314 µg N m$^{-2}$ day$^{-1}$ (median; range 26–3090), which is comparable to the $N_2O$ fluxes from the thawed and revegetated Yedoma in Kurungnakh, and by far higher than the fluxes from permafrost-affected soils generally[6].

Of all $N_2O$ fluxes measured at the both study sites, 23% were negative, with highest uptake rates < −25 µg N m$^{-2}$ day$^{-1}$ observed from bare Yedoma sites with high soil water content (WFPS = 67–100%; Fig. 2a). Uptake of atmospheric $N_2O$ is commonly observed in wetland soils, where energetically more

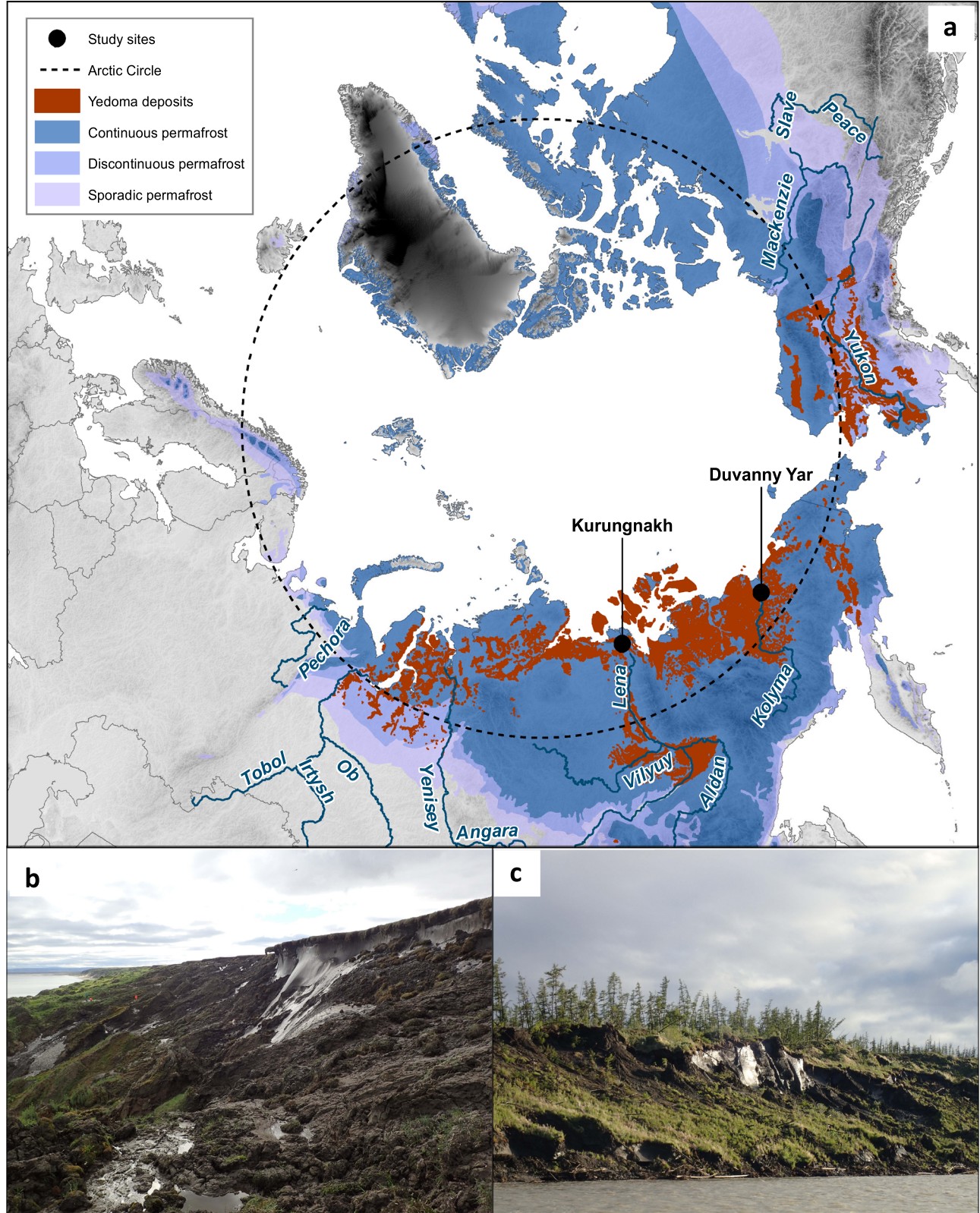

**Fig. 1 Overview of the studied Yedoma exposures. a** Location of the study sites, overlain on the map showing the extent of Yedoma deposits on the Northern Hemisphere[79] and the permafrost zonation[80]. **b** Kurungnakh exposure. **c** Duvanny Yar exposure. Photos b and c by J. Kerttula.

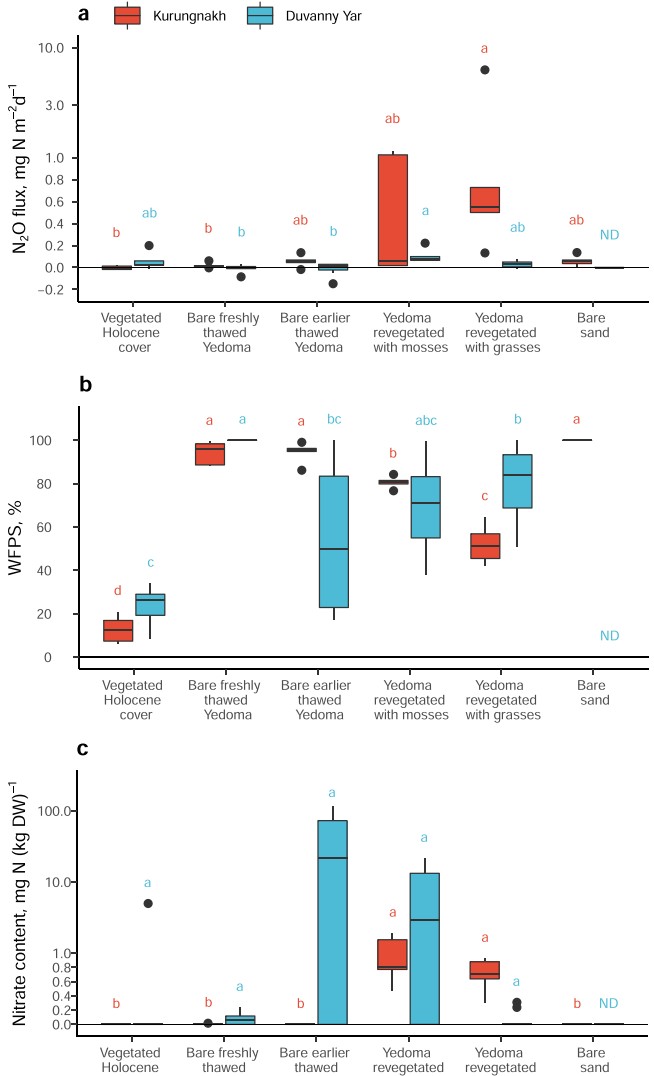

**Fig. 2 Nitrous oxide fluxes and nitrate content at the Kurungnakh and Duvanny Yar exposures. a** In situ $N_2O$ fluxes measured with the chamber technique. **b** Soil moisture expressed as water-filled pore space. **c** Extractable nitrate content. See Supplementary Table 2 for extractable ammonium content. Box plots show lower and upper quartiles, median (thick black line), smallest and largest values without outliers (thin black line) and outliers (circles); $n = 5$ biologically independent samples, except for 'Bare earlier thawed Yedoma' and 'Yedoma revegetated with grasses' in Duvanny Yar, where $n = 10$. Lower case letters indicate significant differences between studied soils, tested separately for each study site (Kruskal-Wallis test with pairwise comparisons with Dunn's test; $p < 0.05$). For $N_2O$ fluxes in a, positive values indicate emissions, and negative values indicate uptake. Note the logarithmic scale on y-axes in **a** and **c**. WFPS water-filled pore space, DW dry weight, ND Not determined.

favorable electron acceptors, such as $O_2$ or $NO_3^-$ are absent[6,23]. However, we occasionally measured small negative $N_2O$ fluxes also from vegetated Holocene cover with very dry topsoil (Fig. 2a; WFPS = 6–15%), supporting previous observations of $N_2O$ uptake in dry oxic soils[24,25]. From all studied microsites, the median $N_2O$ flux was negative only in vegetated Holocene cover in Kurungnakh ($-4\,\mu g\,N\,m^{-2}\,day^{-1}$) and in freshly thawed Yedoma in Duvanny Yar ($-9\,\mu g\,N\,m^{-2}\,day^{-1}$) (Supplementary Table 1).

Nitrous oxide production relies on mineral N supply in excess of the immediate needs of microbes and plants[6,26,27]. High-

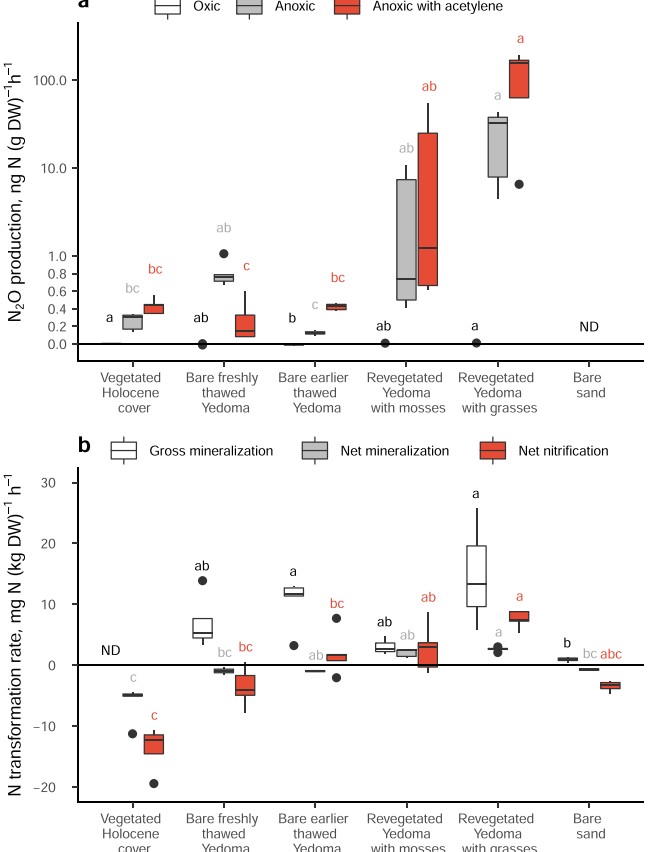

**Fig. 3 Nitrous oxide production and nitrogen transformation rates in Kurungnakh soils. a** Nitrous oxide production with different headspace conditions. Acetylene inhibits the last step of denitrification, $N_2O$ reduction to $N_2$, and can be used to estimate the total denitrification rate. **b** Nitrogen transformation rates including gross N mineralization, net N mineralization and net nitrification. Net N mineralization and nitrification rates were determined with initial N addition (2.1–2.6 mg N $(kg\,DW)^{-1}$) due to low inherent mineral N content in part of the soils. Box plots show lower and upper quartiles, median (thick black line), smallest and largest values without outliers (thin black line) and outliers (circles); $n = 5$ biologically independent samples. Lower case letters indicate significant differences between studied soils, tested separately for each treatment (**a**) or process (**b**; Kruskal-Wallis test with pairwise comparisons with Dunn's test; adjusted $p < 0.05$). Note the logarithmic scale on y-axis in **a**. One outlying point has been removed from net nitrification data for vegetated Holocene cover in **b**. DW dry weight, ND Not determined.

latitude soils usually have a very low content of mineral N species, particularly of nitrate ($NO_3^-$)[28], which can be expected to limit $N_2O$ emissions. At the Kurungnakh exposure, only the high-emitting revegetated Yedoma surfaces had measurable $NO_3^-$ content, while no $NO_3^-$ was detected elsewhere (Fig. 2c, Supplementary Table 2). The opposite spatial pattern in ammonium ($NH_4^+$) content (Dunn's test, $p < 0.05$; Supplementary Table 2) could reflect high $NH_4^+$ consumption by nitrification in thawed and revegetated Yedoma (see below). In Duvanny Yar, $NO_3^-$ content was generally low, except in bare and moss-covered thermokarst mounds called baydzherakhs (Supplementary Fig. 2c, e), where high $NO_3^-$ content up to 116 mg N (kg dry weight (DW))$^{-1}$ was found (Fig. 2c). Strong $NO_3^-$ accumulation indicates high nitrification activity in these dry, well-aerated soils, and availability of precursors for $N_2O$ production by anaerobic denitrification in deeper, water-saturated soil layers or after rain events.

**Effect of moisture and vegetation on $N_2O$ fluxes and N transformation rates**. In previous studies, the highest emission rates of $N_2O$ from permafrost-affected soils have been found from soils without living plant cover, such as bare peat surfaces of permafrost peatlands or retrogressive thaw slumps lacking vegetation[6,22,29,30]. The main reason behind the generally higher emissions from unvegetated than vegetated soils is obvious: when plants are not taking up N from soil, the reactive N forms are entirely available for microbial activities and growth, including the microbial N transformations producing $N_2O$[6]. So, why did the highest emissions in this study occur at sites revegetated after thaw, and not from bare parts of the exposure as expected?

An explanation for the high $N_2O$ emissions from revegetated compared to bare Yedoma could be that plant colonization indicates stabilization of the thawing Yedoma slope after the initial stages of rapid degradation and thaw slumping. This stabilization occurs at a time span between years to decades and is coupled with decreased sediment and water input and improved drainage[19]. The high-emitting revegetated Yedoma surfaces on Kurungnakh were located in the middle part of the exposure (Supplementary Fig. 6a) with intermediate soil moisture content (water-filled pore space (WFPS) 42–84%), as well as the high-emitting Yedoma surfaces revegetated with mosses in Duvanny Yar (Fig. 2a). The revegetated Yedoma surfaces with grasses in Duvanny Yar were located lower down the slope and had higher moisture content (WFPS 69–90%). The role of soil moisture as a primary environmental control on $N_2O$ fluxes, and the bell-shaped dependence of $N_2O$ fluxes on soil moisture peaking at the intermediate soil moisture range are well-documented[6,27]. At intermediate soil moisture content, both oxic and anoxic microsites coexist, providing suitable environments for the two main microbial processes responsible for $N_2O$ production in soils: aerobic nitrification (oxidation of $NH_4^+$ via nitrite ($NO_2^-$) to $NO_3^-$, $N_2O$ as by-product) and anaerobic denitrification (reduction of $NO_3^-$ and $NO_2^-$ to gaseous N forms NO, $N_2O$ and $N_2$)[31].

Intermediate soil moisture content is also optimal for N mineralization (N release from organic matter as a result of microbial decomposition), which is supressed by very wet or very dry soil conditions[7]. At permafrost thaw sites, liberation of mineral N species from permafrost directly at thaw enhances N availability in the short-term[32,33], but in the long-term, post-thaw N mineralization is a more important mechanism of mineral N supply[34]. While there was no correlation between $NH_4^+$ content and $N_2O$ flux and only a weak positive correlation between $NO_3^-$ content and $N_2O$ flux ($R = 0.25$, $p < 0.05$, $n = 30$), we found strong positive correlations between $N_2O$ emissions and net N mineralization ($R = 0.68$, $p < 0.001$, $n = 30$) and net nitrification rates ($R = 0.68$, $p < 0.001$, $n = 30$). The improved drainage and associated enhancement of nitrification ($NO_3^-$ supply) was likely an important trigger for the substantial $N_2O$ release from post-thaw Yedoma.

On Kurungnakh the net N mineralization rates were higher in Yedoma revegetated with grasses than in bare freshly thawed Yedoma (Fig. 3b, Supplementary Table 3). The negative net N mineralization, i.e., net N immobilization, in freshly thawed Yedoma can be explained by high uptake of mineral N species into microbial biomass, exceeding the rate of N liberated from organic matter. In contrast, high net N transformation rates in revegetated Yedoma indicate that the microbial needs for mineral N are well met as a result of continued mineralization after thaw, which allows $N_2O$ emissions to occur even in the presence of plant N uptake. Even stronger net immobilization than in freshly thawed Yedoma was found in vegetated Holocene cover (Fig. 3b, Supplementary Table 3), suggesting limited N mineralization in this dry soil (WFPS 13 ± 6 %; Supplementary Table 1) with a high

C/N ratio (38 vs. 14–15 in Yedoma; Supplementary Table 2)[35]. The increasing trend with post-thaw age was even stronger for net nitrification than for net mineralization (Fig. 3b, Supplementary Table 3).

Despite the fact that plants can effectively compete for N species with soil microbes and suppress N losses by inhibition of nitrification and denitrification processes[36], plants may also promote soil N cycling processes. Rhizosphere priming[37–39] is the term used for the summed effects of different mechanisms by which plants enhance SOM decomposition. These mechanisms include rhizospheric deposition of labile carbon compounds, which provide an easily available source of energy and C, and organic acids, which help to release protective organic-mineral associations, as well as the effect of roots on soil aggregation. Positive priming of N mineralization has been previously reported in high-latitude ecosystems[40,41] and in nutrient-rich croplands grown with perennial grasses[42]. In our study, there was a tendency towards higher gross N mineralization rate in Yedoma revegetated with grasses compared to freshly thawed Yedoma (Fig. 3b). Also, the positive correlations between $N_2O$ emissions with C content ($R = 0.52$, $p = 0.05$, $n = 15$), N content ($R = 0.60$, $p = 0.02$, $n = 15$) and $CO_2$ fluxes (ecosystem respiration; $R = 0.41$, $p = 0.002$, $n = 55$) in the optimal moisture range of WFPS 45–85% suggest that plant-derived organics might stimulate N cycling processes at the Kurungnakh exposure. Additionally, grasses may have caused changes in the soil porosity and macropore structure that favor $N_2O$ production[43,44]. However, it is difficult to separate the plant effects on $N_2O$ emissions from the effects of moisture changes. Similarly, the increase in ecosystem respiration rates with slope stabilization and revegetation was observed in both exposures (Supplementary Fig. 6b, Supplementary Table 1), reflecting the joint effect of drying and increased plant C input, which cannot be distinguished from each other at this stage.

**Changes in microbial community composition related to N cycling and $N_2O$ production**. By using a targeted metagenomics tool designed to capture the genes responsible for key functions of the N cycle (Ref. [45]; see Methods), we here reveal another important mechanism driving the increase in $N_2O$ emissions with time after thaw: changes in microbial community composition. We observed significant changes across the Yedoma exposure in the relative abundance of nitrification and denitrification genes with time passed after thaw, associated drainage and plant colonization (Fig. 4, Supplementary Fig. 8). These changes occurred within just a couple of years and led to strikingly different microbial community structure related to N cycling in thawed Yedoma compared to the Holocene cover deposits that feature well-developed cryosols prevailing in the region. The sparse studies that have previously reported changes in N cycling genes associated with permafrost thaw are in line with our findings here: Significant increases in denitrification genes (norB, nirS, nosZ) have been observed following thaw[30,46].

In detail, we found an increase in the relative abundance of the amoA gene (first step of nitrification) from all the captured genes from 0.6% in freshly thawed Yedoma to 2.5–3.5% in revegetated Yedoma surfaces (Fig. 4). These results were supported by increasing copy numbers of bacterial and total (archaeal + bacterial) amoA gene with time after thaw (Supplementary Fig. 7). At the same time, the proportion of the nir genes (nirK + nirS) contributing to $N_2O$ production doubled from 15% to 29% in Yedoma revegetated with grasses, which was further coupled with halved proportion of the nosZ gene (catalyzing the reduction of $N_2O$ into $N_2$; Fig. 4). These opposite trends in nir and nosZ genes resulted in a significant increase of the

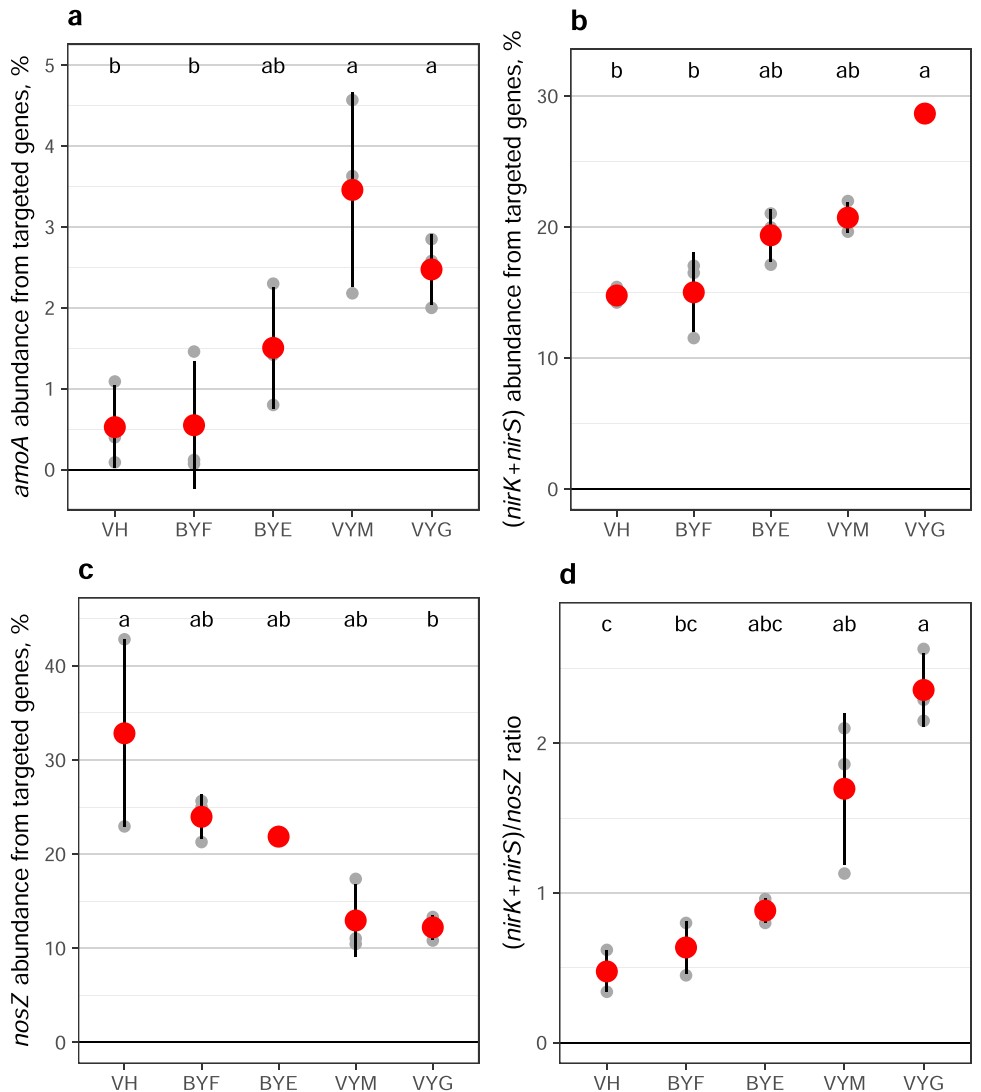

**Fig. 4 Relative abundance of selected N cycling genes at the Kurungnakh exposure from all functional gene sequences captured with the targeted metagenomics tool. a** Relative abundance of *amoA* gene (including bacterial and archaeal). **b** Relative abundance of *nir* gene (including both *nirK* and *nirS*). **c** Relative abundance of *nosZ* gene. **d** Ratio of (*nirK* + *nirS*)/*nosZ* genes. The studied surfaces are arranged according to the distance from the Yedoma cliff border, with intact Holocene cover on the top of the Yedoma exposure on the left and earliest thawed revegetated Yedoma on the right side. Small gray symbols indicate values for individual samples, large red symbols indicate means, and error bars indicate standard error of mean (*n* = 3 biologically independent samples). Lower case letters indicate significant differences between studied soils (Kruskal-Wallis test with pairwise comparisons with Dunn's test; unadjusted *p* < 0.05). VH Vegetated Holocene cover, BYF Bare freshly thawed Yedoma, BYE Bare earlier thawed Yedoma, VYM Yedoma revegetated with mosses, and VYG Yedoma revegetated with grasses.

(*nirK* + *nirS*)/*nosZ* ratio, a commonly used indicator of $N_2O$ production potential in soils (Fig. 4), which was shown to increase with permafrost thaw in mineral upland soils[30]. Vegetated Holocene cover had the lowest relative abundances of *amoA* and *nir*, and the highest relative abundance of *nosZ* among all studied soils, which together explain the low $N_2O$ emissions there.

In addition to the above-mentioned denitrification genes, also the relative abundance of the *nrfA* gene encoding nitrite reduction to ammonia was very low in vegetated Holocene cover (0.4%), intermediate in bare Yedoma (1.7–1.8%) and highest in revegetated Yedoma (2.0–3.2%; Supplementary Fig. 9). The *nrfA* gene is a key functional gene in dissimilatory $NO_3^-$ reduction to ammonium (DNRA), and its increasing abundance with post-thaw age reflects the improved availability of $NO_2^-$ from nitrification. The *norB* gene responsible for nitric oxide (NO) reduction to $N_2O$ did not show similar gradual increase from bare

to revegetated Yedoma sites as other nitrification and denitrification genes (Supplementary Fig. 9). This might be a result of a methodological bias: since *norB* has been less studied than *nir* and *nosZ*[47], the gene databases used for developing the probe capture tool are not including enough probe diversity for *norB* to cover Arctic variants. Also, the nitric oxide reductase encoded by *norB* has a role in detoxification of NO, giving this enzyme a broader importance than just catalyzing an intermediate step of the denitrification pathway[48].

To test whether the low $N_2O$ production in bare freshly thawed Yedoma was not merely a consequence of the high water-saturation, we dried the soil (25% reduction in the water content) and repeated the incubation under oxic and anoxic conditions with and without acetylene addition (see Methods). Drying indeed caused a 6-fold increase in $N_2O$ production under oxic treatment from the initial very low production rate (Wilcoxon signed-rank test, *p* = 0.03; Supplementary Fig. 10). Under anoxic

treatment, drying with and without C addition even reduced the $N_2O$ production ($p = 0.01$). But, when we amended the soil with $NO_3^-$ in addition to C, the $N_2O$ production increased drastically in all three headspace treatments (725, 12 and 379-fold in oxic treatment, anoxic treatment and anoxic treatment with acetylene, respectively; $p = 0.01$). This shows that even after creating favorable conditions for nitrification by drying, $N_2O$ production in anoxic conditions was still limited by $NO_3^-$ due to the low abundance of ammonia oxidizers. Taken together, our results represent clear in situ evidence for the microbial limitation of N cycle and $N_2O$ emissions from thawing Yedoma permafrost due to low abundance of ammonia oxidizers, confirming the findings of a recent laboratory incubation study, which discovered this phenomenon[49].

Similarly to ammonia oxidizers in the present study and in the earlier laboratory study[49], it has been shown previously that also methanogens represent a bottle-neck in Yedoma permafrost biogeochemistry:[50] a small part of the microbial community carrying out an important function associated to permafrost-climate feedbacks. On the studied Yedoma exposures, the low $CH_4$ emissions from bare freshly thawed Yedoma (<0.5 mg C m$^{-2}$ day$^{-1}$) despite the high water saturation suggest that also there methane production was limited by lack of methanogenic archaea (Supplementary Fig. 6, Supplementary Table 1). Most of the studied soils showed minor $CH_4$ emission or consumption rates (Supplementary Table 1), and even the highest median $CH_4$ emission 1.95 mg C m$^{-2}$ day$^{-1}$, observed in revegetated yedoma with mosses in Kurungnakh, was modest compared to the emissions from polygonal wetlands (2–35 mg C m$^{-2}$ day$^{-1}$)[51] and small ponds (4–35 mg C m$^{-2}$ day$^{-1}$)[52] in the same region. Lack of recovery of methanogenic function with post-thaw age is easy to explain with drying associated with slope stabilization at such Yedoma exposures[19], which creates unfavorable conditions for anaerobic methanogens.

At the same time, our results demonstrate that although $N_2O$ production in recently thawed Yedoma permafrost is restricted by the microbial community composition, retrogressive thaw slumps provide ideal conditions for the development of active $N_2O$ producing microbial community, leading to high $N_2O$ release within less than a decade. This highlights that short-term laboratory experiments indicating microbial limitations in the C and N cycles of permafrost soils [49,50] do not well represent the real changes in microbial community and their functioning with time.

**N losses from thawing Yedoma permafrost and their implications**. We show here that thawing Yedoma exposures host sites with optimal conditions for intense microbial N cycling and associated $N_2O$ production. Although $N_2O$ emissions may decrease with further slope stabilization as a result of continuous N losses and establishment of full vegetation cover[30], the retrogressive thaw at the same time keeps releasing fresh sediments, rich in N available for microbial activities. According to our remote sensing analysis using ArcticDEM and UAV data, the Yedoma cliff of Kurungnakh retreated in 2012–2019 as a result of permafrost thaw at a rate of 3.7 (2.5–5.7) m year$^{-1}$ (median with (25th–75th percentiles); Supplementary Figs. 3 & 4; see Methods section for details). Based on typical ice content of Yedoma deposits and the total N content of freshly thawed permafrost, we could estimate that this retrogressive thaw liberated at the thaw front as much as 1.7 kg of total N per m$^2$ per year, which was associated with a release of 39 g of mineral N per m$^2$ per year (see Methods section for details). These are remarkably high amounts of added N compared to the main pathways of external N input in high-latitude ecosystems: biological N fixation of 20–200 mg N

m$^{-2}$ year$^{-1}$ [53] and atmospheric N deposition of <200–300 mg N m$^{-2}$ year$^{-1}$ [54]. The additional N from Yedoma permafrost will have important consequences for plant growth and associated C fixation[55], lateral N losses to waterbodies[20,56] and gaseous N losses to the atmosphere[34], and importantly $N_2O$ fluxes, as shown here.

In parallel to the retrogressive thaw front, the zone with optimal conditions for high $N_2O$ emissions on the middle part of the slope will likely shift spatially but persist as an active zone along this retreating Yedoma shore. Based on the emissions from disturbed Yedoma revegetated with grasses on Kurungnakh (median multiplied with a snow-free-season-length of 100 days), we estimated that under these optimal conditions thawed Yedoma will lose 54.8 mg N m$^{-2}$ of $N_2O$ to the atmosphere just in one year. This corresponds to 0.14% of the mineral N originally liberated at the permafrost thaw front from a similar area in a year (see above). This is seven times lower than the IPCC $N_2O$ emission factor for N fertilization in managed mineral soils (1%)[57], but still high considering that it occurs in a pristine northern soil, which are generally N limited[28].

While it is important to remember that such high $N_2O$ emissions will occur in particular settings (Yedoma exposed to surface, suitable moisture content, sufficient time after thaw for establishment of $N_2O$ producing microbial community), these conditions are not limited to the retrogressive thaw slumps along rivers studied here. Similar disturbed N-rich Yedoma with successional plant cover are widespread along thermokarst lake shores, coasts, slopes, and valleys across the Yedoma region (Supplementary Fig. 11). Widespread occurrence of such land-forms suggests that our findings are the first indication for substantial $N_2O$ emissions over large areas in the Arctic. We show that N liberated from this ancient permafrost during thaw is highly available for mineralization and further microbial activities. With rapid Arctic warming and associated permafrost thaw, the huge N resources contained in Yedoma will become increasingly available with important implications on ecosystem functioning and climate feedbacks at local to global scales.

## Methods

**Study sites**. Nitrous oxide fluxes were studied at two study sites located in Northeast Siberia, Russia: the Kurungnakh exposure (N 72°20', E 126°17'), located on Kurungnakh-Sise Island in the Lena River Delta and the Duvanny Yar exposure (68°38' N, 159°09' E), located by the Kolyma River (Fig. 1). Both study regions are underlain by continuous permafrost, and the climate is continental Arctic with mean annual air temperature of –12.3 °C and –11 °C and annual rainfall amounts to 169 mm in Kurungnakh and 197 mm in Duvanny Yar, respectively[58–60]. More information about the climatic conditions in the region, depositional characteristics and vegetation can be found in Supplementary Note 1 and the references therein. Altogether, the following surfaces types were chosen for the study ($n = 5$–10): 1) vegetated Holocene cover deposits overlying undisturbed Yedoma permafrost on the top of the exposure; 2) freshly thawed Yedoma bare of vegetation, close to thawing ice-wedges in the upper part of the exposure; 3) earlier thawed Yedoma bare of vegetation; 4) disturbed Yedoma in the lower, stabilized parts of the slope, revegetated by mosses; 5) disturbed Yedoma revegetated by grasses; and, only on Kurungnakh, 6) bare sand close to the river shore receiving Yedoma melt-waters by a small stream running through the exposure (Supplementary Figs. 1 & 2).

**In situ $N_2O$ fluxes**. In situ nitrous oxide ($N_2O$) fluxes were measured by the static chamber technique[22], twice in July 2016 on Kurungnakh and once in July 2017 in Duvanny Yar (see Supplementary Note 1). Five gas samples were drawn from the chamber headspace within a 50-minute enclosure time and transferred into pre-evacuated 12 ml glass vials (Labco) for storage until the analysis. Soil temperature and moisture as volumetric water content (VWC) were recorded in the vicinity of the chamber. The $N_2O$ mixing ratios were determined with a gas chromatograph (GC; Agilent 7890B Agilent Technologies, Santa Clara, CA, USA) equipped with an autosampler (Gilson Inc., WI, Middleton, USA), an electron capture detector (ECD) for $N_2O$ and a flame ionization detector (FID) for $CH_4$. Fluxes of $N_2O$ were calculated from the slope of the linear increase of the $N_2O$ mixing ratio in the chamber headspace as a function of time. Besides initial visual inspection, the quality control of gas flux results was based on inspection of Root Mean Square Error (RMSE) in ppm (RMSE > 3 * SD) as compared to the variability of standard

gas mixtures in a similar range. Methane ($CH_4$) fluxes were obtained from the same chamber measurements as the $N_2O$ fluxes. Carbon dioxide ($CO_2$) fluxes in the dark including the plants (ecosystem respiration) were measured with the dynamic chamber technique[61] using an infrared gas analyzer (Li-840, LiCor Lincoln, Nebraska, USA in Kurungnakh; EGM-4, PP Systems, Amesbury, MA, USA in Duvanny Yar).

**Soil sampling and analysis**. Soil samples were taken from the topsoil (0–10 cm), cleaned from stones and roots and homogenized by sieving (mineral soils; 5 mm mesh size) or by hand-mixing (organic soils). Bulk density was determined from volumetric soil samples after drying until constant weight at 60 or 105 °C for organic and mineral soils, respectively. Particle density was determined by a pycnometric method. Total content of C, organic C and N, as well as $\delta^{13}C$ of SOC and $\delta^{15}N$ in the bulk soil were analyzed with an elemental analyzer (Thermo Finnigan Flash EA 1112 Series, San Jose, CA, USA). For organic C analysis, inorganic C was removed from a subsample with the acid fumigation method[62]. Water-filled pore space (WFPS) was calculated from VWC measured in situ, using bulk density and particle density determined as described above. Soil pH was measured from slurries with a soil:$H_2O$ ratio of 1:4 ratio (w/v). For determination of mineral N content, ammonium ($NH_4^+$) nitrate ($NO_3^-$) were extracted from freshly sampled soils at the field laboratory (1 M KCl, a 1:3 volume ratio of soil to extractant). The extracts were frozen for storage until the analysis by spectrophotometric methods as previously described[61]. See Supplementary Note 1 for further details about the soil analysis.

**Gross and net N transformation rates**. Nitrogen transformation rates were determined in the field laboratory from freshly sampled soils to imitate the processes occurring in the field during the flux measurements as realistically as possible. Due to time limitation and logistical challenges related to the fieldwork in Duvanny Yar, we did not measure N transformation rates from Duvanny Yar soils, but only from soils sampled in our primary study site Kurungnakh. For the determination of gross N mineralization and nitrification rates, we used the pool dilution method, which is based on labeling the product pool ($NH_4^+$ for mineralization, $NO_3^-$ for nitrification) with the heavy N isotope $^{15}N$[63–65]. Due to the low mineral N content and high N immobilization, we were not able to determine gross nitrification in any of the studied soils and gross mineralization in some of the soils. However, even in these cases, we could use the data to calculate the net N mineralization and nitrification rates as described below.

In brief, two sets of samples (2 g of fresh soil) were prepared for both N mineralization and nitrification measurements. We added 500 µl of 0.25 mM, 10 at-% ($^{15}NH_4$)$_2SO_4$ solution to the N mineralization samples, and 500 µl of 0.50 mM, 10 at-% $K^{15}NO_3$ solution for nitrification samples. This N addition amounted to 2.1–2.6 mg N (kg DW)$^{-1}$ depending on the soil moisture content. After labeling, the samples were incubated for 24 h at the approximate in situ temperature of ~5 °C. Nutrient levels ($NO_3^-$ and $NH_4^+$) were determined from samples extracted at two-time points of 4 and 24 h with 2 M KCl as described above. Content of $^{15}N$ in $NH_4^+$ extracts was analyzed by continuous-flow isotope ratio mass spectrometer (IRMS; Thermo Finnigan DELTA XPPlus, San Jose, CA, USA) coupled to an elemental analyzer (Thermo Finnigan Flash EA 1112 Series) and an open split interface (Thermo Finnigan Conflow III) after conversion to solid phase by the microdiffusion method as previously described[22]. Net N mineralization rates were calculated by dividing the difference of the total mineral N content ($NH_4^+$ and $NO_3^-$) between the first and second sampling points with the incubation time. Net ammonification and nitrification rates were calculated similarly from the change in $NH_4^+$ and $NO_3^-$ contents, respectively.

**$N_2O$ production and total denitrification rates in laboratory incubations**

*Experiment 1: Nitrous oxide production under different headspace conditions*. Rates of $N_2O$ production and total denitrification were determined by incubation experiments from soils sampled in Kurungnakh, our primary study site (See above in Gross and net N transformation rates). The soils were kept frozen during storage and shipment to the laboratory, thawed, homogenized by hand, and further stored at 4 °C. After three days of acclimatization at the incubation temperature, the soil samples (10 and 25 g fresh weight for organic and mineral soils, respectively; $n = 5$) were incubated at field moisture content at 10 °C under three different headspace treatments: 1) oxic, 2) anoxic and 3) anoxic with acetylene. For oxic treatment (1), laboratory air was used as headspace. For anoxic treatments with and without acetylene (2 and 3), the flasks were closed inside a glove bag after flushing several times with $N_2$ gas (purity ≥ 99.999%). Acetylene was added into the third treatment at 10 vol-% to block the last step of denitrification, reduction of $N_2O$ to $N_2$, thus making $N_2O$ as the final denitrification product[66]. Gas samples were taken at five-time points at days 0, 1, 2, 3, and 6, and analysed for $N_2O$ mixing ratios with GC as described above. For oxic treatment, flux per mass of dry soil was calculated along the slope of a linear regression fitted to the first four sampling points with constant $N_2O$ production rate. For the anoxic treatments, we report the maximum $N_2O$ production between two sampling points, because we often observed $N_2O$ consumption (treatment 2 without acetylene) or steady state (treatment 3 with acetylene) after initial $N_2O$ production, indicating reduction of $N_2O$ to $N_2$. See Supplementary Note 1 for more details about the incubation experiments.

*Experiment 2: Response of nitrous oxide production to different moisture conditions and carbon and nitrogen sources*. The aim of the second incubation experiment was to investigate the factors limiting $N_2O$ production in freshly thawed Yedoma. We dried the freshly thawed Yedoma to reduce the water content by 25%, weighed 17 g FW to incubation flasks, and incubated under three different headspace treatments as described above. For each headspace treatment, we applied three different amendments within a volume of 250 µl per flask: control (milli-Q $H_2O$ addition), addition of C (glucose; 67 µg C (g DW)$^{-1}$, equal to 0.3% of SOC), or addition of C (as above) and $NO_3^-$ (4.7 µg N (g DW)$^{-1}$, equal to 0.3% of TN). The GC analysis and calculation followed the procedure described above for Experiment 1.

**Molecular studies on microbial community participating in N cycling**. For molecular studies, we sampled five surface types in the primary study site Kurungnakh ($n = 3$). The studied surfaces represented different stages of thermokarst and post-thaw succession: vegetated Holocene cover, freshly or earlier thawed Yedoma, revegetated Yedoma with mosses or with grasses. We extracted DNA from these samples in three technical replicates as described previously[67], see Supplementary Note 1 for details.

Quantitative PCR (qPCR) of archaeal and bacterial *amoA* and *16S rRNA* genes was performed using the reaction and cycling conditions as described previously and summarized in Supplementary Table 5[67–70]. All reactions were performed in duplicates. The specificity of qPCR amplification products was verified by melting-curve analysis and gel electrophoresis.

To detect the changes in N cycling-relevant microbial community structure with permafrost thaw and post-thaw succession, we studied the relative abundances of the key N cycling genes using a captured metagenomic tool. The method has been validated and tested for overall performance and specificity of the probes used for sequence capture[45], and it has been successfully applied for studying N cycling genes in bioreactors treating aquaculture effluents[71]. This method is designed for targeting and sequencing the organisms carrying the key N cycling genes involved in the following processes: $N_2$ fixation (*nifH*), nitrification (*amoA*), $NO_3^-$ reduction (*narG*, *napA*), denitrification (*nir* (*nirK* + *nirS*), *norB*, *nosZ*), dissimilatory nitrate reduction to ammonium (DNRA) (*nrfA*) and anammox (*hdhA*) using gene-specific probes following the NimbleGen SeqCap EZ protocol by Roche NimbleGen, Inc. A detailed description of the method can be found in Supplementary Note 1.

**Statistical analysis**. All statistical analyses were conducted with R, version 3.6.1[72]. We used histograms, Q-Q plots and the Shapiro-Wilk normality test for testing normal distribution of the data. Differences in $N_2O$ fluxes, soil physicochemical characteristics and N transformation rates between the surface types were tested separately within each study site (fluxes, soil characteristics) or for each treatment ($N_2O$ production in soil incubations). For non-normally distributed data, we used the non-parametric Kruskal–Wallis test followed by pairwise comparisons with Dunn's test of the *FSA* package[73]. For normally distributed data, we used Welch's one-way analysis of variance (ANOVA) followed by pairwise comparisons with the Games-Howell post hoc test of the *userfriendlyscience* package[74]. The equality of variances prior to ANOVA was tested with the Bartlett's test. For testing the differences in molecular data (relative abundances, copy numbers & gene ratios; $n = 3$) between the studied soils the Kruskal–Wallis test with Dunn's post hoc tests was also used. Treatment differences in the incubation experiments were tested with Wilcoxon signed-rank test. The role of soil characteristics, mineral N content and N transformation rates as drivers of in situ $N_2O$ fluxes was explored by the non-parametric Spearman correlation analysis with the *Hmisc* package[75].

**Rate and volume of thermal erosion and related N mobilization at the Kurungnakh exposure**. We estimated the rate and volume of thermal erosion for the south-eastern part of the riverbank on Kurungnakh Island (length of the section 1.7 km) using Arctic DEM datasets from March 2012 (WorldView-1/WorldView-1 imagery) and April 2014 (WorldView-2/WorldView-2 imagery)[76] and a digital elevation model (DEM) from unmanned aerial vehicle (UAV) imaging from July 2019. The analysis was done in QGIS v 3.14 (with SAGA and GRASS). Detailed description of the method is available in Supplementary Note 1.

The calculation of the Yedoma retreat between 2012 and 2014 and 2014 and 2019 was based on the following steps: 1) Delineating the shoreline in 2019 based on digital elevation model (DEM) from UAV imaging in 2019; 2) delineating the Yedoma cliff boundary on the top of the exposure based on the DEM and orthophoto mosaic from UAV imaging in 2019, and 3) calculating distances between the shoreline and the cliff boundary 2019 and building intersection points for these lines with the cliff boundary determined from Arctic DEM datasets from March 2012 and April 2014. This algorithm resembles procedures that are available from the free AMBUR package[77]. Between 2012 and 2019 the Yedoma boundary retreated as a result of thermal erosion by 3.7 (2.5–5.7) m year$^{-1}$ (median with 25th–75th percentiles). Then, we calculated the volume of eroded material along the studied cliff boundary from DEM difference (DEM-2019 subtracted from ArcticDEM-2014 and ArcticDEM-2012; the resulting value, if positive, means subsidence due to thaw and erosion). For this calculation, we used the 'Raster surface volume instrument' in QGIS. The resulting volume estimate is 223,502 m$^3$.

We further approximated how much total and mineral N was annually liberated at the Kurungnakh exposure as a result of permafrost using the annually eroded

volume per unit area, the total ground-ice content of 82 vol %[14], and bulk density of 1.22 g cm$^{-3}$, total N content of 0.16% and mineral N content of 35.3 mg N (kg DW)$^{-1}$ detected in freshly thawed Yedoma in this study (Supplementary Table 2). This resulted in total N release from thawing permafrost at the Kurungnakh cliff boundary of 1.7 kg N m$^{-2}$ year$^{-1}$ and mineral N release as $NH_4^+$ of 39 g N m$^{-2}$ year$^{-1}$.

## Data availability
Processed data files supporting the findings can be accessed in Zenodo, https://zenodo.org/record/5360646[78]. The metagenomic data are deposited to the SRA database under the BioProject link PRJNA771879.

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

## Acknowledgements

We wish to acknowledge funding from the Academy of Finland in the frame of the projects Yedoma-N (no. 287469), NOCA (no. 314630) and the Atmosphere and Climate Competence Center (ACCC) (no. 337550). The work of M.E.M was additionally supported by PANDA (no. 317054), C.V. by MUFFIN (no. 332196) and H.M.P.S by AMOBORA (no. 290315) and NITRIOBIOME (no. 342362), all funded by the Academy of Finland. A.F. and N.L. by Russian Foundation for Basic Research project "Assessing the Circumpolar Balance of N₂O" (18-55-11003 AF_t), C.K. by the BMBF project CarboPerm (no. 03G0836A) and G.G. and A.M. by BMBF KoPf Synthesis (03F0834B). Field support in the Lena Delta was provided in the framework of the joint Russian-German LENA expeditions. We are thankful for the personnel of the Research Station "Samoylov Island" and its operator - IPGG SB RAS, as well as for the personnel of the Northeast Science Station for invaluable support during the fieldwork. We acknowledge T. Trubnikova for important contribution to field logistics and laboratory assistance, N. Welti for fieldwork preparations, E.J. Wilcox for help in preparing Fig. 1., and C. Palacin-Lizarbe for valuable discussions.

## Author contributions

C.B., M.E.M., K.D. and P.J.M designed the study; M.E.M., C.V., J.K., C.B., C.K., A.F. and N.L. conducted the fieldwork; A.M., G.G., C.K., N.Z. and S.Z. provided access to and expertize on the study sites and supported the project with field logistics; M.N., K.D. and M.E.M. conducted the incubation experiment; J.K., J.G.R. and H.M.P.S conducted the molecular studies; A.F. was responsible for the remote sensing analysis; M.E.M., J.K., M.N., K.D., J.G., L.V.D. and C.B. conducted other laboratory analysis and data processing; M.E.M. wrote the first version of the manuscript with contribution from J.K., after which all co-authors provided input on manuscript text, figures, and discussion of scientific content.

## Competing interests

The author declares no competing interests.
