## [Peer Review File · Nature Communications]

REVIEWER COMMENTS

Reviewer #1 (Remarks to the Author):

Permafrost affected soils are well recognized sources of the greenhouse gas methane and highly vulnerable to climate change. Nitrous oxide emissions were traditionally considered neglectable. Such a view is changing during the last decade, and Marushchak et al. nicely demonstrate that certain 'hot spots' in the permafrost zone are important factors to be considered when it comes to modelling of climate change. The present study is indeed timely, well conceived, highly interesting and nicely demonstrates in situ nitrous oxide emissions from vegetated, aged Yedoma permafrost. Even more important is the demonstrated potential of the sampled soils to produce large amounts of nitrous oxide when the conditions become more favourable for the processes involved, i.e., nitrification and denitrification. The study addresses parameters affecting in situ nitrous oxide emission along with in depth microbiological analyses of associated organisms. During the presentation of the data, information on key microbes associated with nitrous oxide emissions was only touched upon. I would encourage the authors to provide more information on the relevant functional gene diversity obtained, potential key taxa and deposit sequencing data (provide accession numbers), if not already done. To provide figures and tables on the microbiological data in the supplementary materials is obviously an option. Taken together, this well conceived study with attention to detail has a high degree of novelty and represents a major step forward.

Specific comments:

L1 and throughout the manuscript. Shouldn't "Yedoma" and other proper nouns start with a capital?

L34 Please avoid first time claims. The study provides obviously highly novel data; otherwise it shouldn't be published ...

L92/L118 Negative nitrous oxide emission rates indicate that such soils might represent a sink for nitrous oxide under special conditions. I would encourage to evaluate this finding (development stage, vegetation, soil parameters, microbiome, etc.) and add a short statement on this topic.

L97/98 repetition of "already"; please avoid.

L128 What is the proportion of vegetated Yedoma relative to total Yedoma? Is there any information on how long such "hot spots" of nitrous oxide emissions remain stable?

L137 Shouldn't "Baydzherakhs" start with a capital letter?

L213 Should this read: different microbial community structure related to N-cycling ?

Furthermore, some more details on the microbial community structure (see comment above) and discussion in context of microbiological literature on nitrous oxide production associated permafrost microbiomes would fit here and after the next paragraph.

L216-227 What about the nor genes (same question to Fig. 4)? Nitrous oxide is the end product of Nor rather than Nir. Did the nor genes follow a similar trend like the nir genes?

L240 Duplicate "the".

L458 Duplicate "S1".

L503 Fragmented sentence "We 500 µl ...".

L518 Please provide an explanation here, why only Kurungnakh soils were utilised for incubation experiments.

L533 Duplication of "constant".

L555 Shouldn't this read Table S5? Please check all references to data sets carefully.

L565 Which nir genes were included? nirK and nirS? Please clarify.

Figure 2 b. "100.0" etc. One digit might suffice here ("100" etc.). Please define ND in the legend.

Reviewer #2 (Remarks to the Author):

Comments on the manuscript of Marushchak et al submitted to Nature Communications

The manuscript presents the results of a study on soil N cycling and N₂O emissions in two sites of Yedoma permafrost soils. They elegantly show, along a gradient of permafrost thawing, that nitrification and denitrification processes, which were absent in recently thawed permafrost, are slowly activated with soil exposure to surface microbes and plants. This reactivation is associated to large N₂O emissions, a power greenhouse gas. Their results confirm a recently published study by Monteux et al. demonstrating the microbial limitations of soil N cycling in Yedoma permafrost using an

inoculation approach. These last convergent news from the scientific community represent a major advance in our understanding on interactions between global warming, N cycling in arctic terrestrial ecosystems and feedbacks on climate. Therefore I fully support the publication of this study. Below some important comments that must be accounted before publication.

L34/35 the first time is incorrect since you confirm the findings of Monteux et al. Moreover, I think the message you carry out is sufficiently important to avoid such artifices. See below advices to how better position your studies regarding to the previous study.

L56/57 This sentence is tendentious and dangerous in term of message to the society (if natural ecosystems are an important source of GHG, why should we protect them and make effort to decrease anthropogenic GHG). Natural ecosystems and agricultural soils simply do not occupy same surface.

L94-98 Could you refer to the position on the slope (top -> down) when you describe your gradient. That will help the understading for non-specialists.

L179-190 I do not understand you invest of most your space to describe N mineralisation/immobilisation along the gradient instead of net nitrification (one line only for this process). Indeed, the change in nitrification along the gradient is higher than that of N mineralisation and this former variable is the most relevant regarding the message you convey (reactivation of nitrification and denitrification).

L193-194 The use of terms "rhizosphere priming effect would be more adapted" in your case. The acceleration is not only caused by energy-C. This rhizosphere priming effect is the consequence of root activities including rhizodeposition, nutrient uptake, soil aggregates disturbance... You should cite the study of Henneron et al 2020 in New Phyt who present different rhizosphere priming and impacts on soil N fluxes induced by perennial grasses, which may explain the pattern observed.

L240-242 This is not correct, this is not the first evidence of the microbial limitation of N cycling in permafrost. A better positioning is needed here to highlight the novelty of your work: your findings support the conclusion of Monteux et al. on the microbial limitation of soil N cycling Yedoma permafrost obtained in an incubation study (in your case it's even not a demonstration, but a correlation). Moreover, you demonstrate in situ (for the first time, if you want to specify that) that this limitation can be levered in some years when thawed yedoma permafrost are exposed to surface microbes and plants. To my opinion this last finding is highly important for societal aspects.

L257-260 despite several reading I did not understand this sentence which are cut too many times by informations included in brackets.

L281 please correct in line with my previous comments.

Reviewer #3 (Remarks to the Author):

Thawing yedoma permafrost as a neglected nitrous oxide source

Marushchak M.E., Kerttula J., Diáková K., Faguet A.4, Gil J.1,5, Grosse 4 G.6,7, Knoblauch, C.8,9, Lashchinskiy N.4,10, Nykamb M.1, Martikainen P.J.1, 5 Morgenstern A.6, Siljanen H.M.P.1,11, van Delden L.1,6, Voigt C.1,12, Zimov N.13, 6 Zimov S.13, Biasi C.1

Comments to the authors

This paper, entitled 'Thawing Yedoma permafrost as a neglected nitrous oxide source' by Marushchak et al. addresses the complex changes in soil process occurrence and -rates in thaw gradients of Yedoma permafrost soil. N₂O is a greenhouse gas with ~300 times the global warming potential of CO₂, the emission of which has been wrongly assumed negligible in permafrost areas (Voigt et al 2020, Nature reviews Earth & Env.). This assumption, combined with the logistically challenging fieldwork, has led to scarce N₂O field data availability for the permafrost area. The field site selected here is representative for Yedoma soils, which contain large carbon (C) and nitrogen (N) stocks that have been immobilized (frozen) for thousands of years. With the biotic awakening of the arctic, these stocks of C and N now start to play their role in the dynamic part of global elemental cycling, and are as such likely a key-player in the permafrost-climate change feedback. The current study presents both measured N₂O emissions from relevant field sites as well as data essential to improve our understanding of the dynamic drivers behind N₂O emissions (mineralization, nitrification, denitrification, soil moisture, microbial community composition) in a changing climate. The authors also provide an estimate of the rate of overall N liberation from Yedoma thaw slumps, a first of its kind as far as I am aware. Moreover, the authors have fitted their experimental design to be able to test, in a field setting, new and exciting hypotheses thus far only based on laboratory incubations (nitrification in freshly thawed permafrost soils hampered by functional limitation of the microbial community, Monteux et al 2020, Nature geosci).

This study provides important field data on both the magnitude of N₂O emission (133–6286 µg N m⁻² day⁻¹) and, more general, N-liberation (1.7 kg m⁻²yr⁻¹) from thawing Yedoma permafrost soils, as well as of the (short, < 10 years) time it takes for 'functionally limited'/'virgin' freshly thawed permafrost soils to become colonized by external microbes. These data are a strong indication that potential estimates of N₂O emission based on laboratory incubations using 'freshly thawed' permafrost soils without inoculation with external microbes might underestimate actual emissions in the field. The non-linear processes shown here are currently not taken into account by Earth System Models and this study will give direction to ongoing discussions between field-researchers and EMS modellers. I find this paper well-written, the data presented in the main text excellent and the supplementary material information all of interest. I did not find flaws in data-analysis, interpretation and conclusions. This study seems to be carefully performed by a brave team of researchers, who took the important and challenging work upon them to get these data out. The authors are not overstating the importance of their work, and they put their data in appropriate perspective.

Minor comments:

l. 483 'bluk soil' -> bulk soil

l. 228 " to demonstrate that", please rephrase to (something like) " to test whether..."

l. 242 In reference 41 and 42 functional limitation of freshly thawed permafrost soil is discussed, and nitrification, (part of) respiration, and methanogenesis are hypothesized as potentially missing functions. You specifically test for and discuss nitrification. But you also seem to have CH₄ and CO₂ data for the same thaw gradient on which you base your estimate of 'colonization time'(my words) by external microbes (i.e. regaining soil functioning). I understand that it might be beyond the scope of this paper, but since you have the results, would it be possible to discuss your CH₄ and CO₂ results shortly, either in the main text or in a short supplementary discussion? I.e. are your findings in line with what was previously observed/hypothesized and if not, what are your thoughts on why?

General:

- Perhaps address shortly in the supplementary methods attention points related to selecting a time and a place for N₂O measurements (hot spots, hot moments) and potential implications.
- Data stored at zenodo ([https://zenodo.org/doi:10.5281/zenodo.4587235](https://zenodo.org/doi/10.5281/zenodo.4587235)) are not actually available for reviewing – it is needed to apply for access from the authors. This is perhaps too time-consuming and, why not, awkward, to be useful for reviewing purposes?

Response to reviewer comments for the manuscript NCOMMS-21-09918-T, titled “Thawing Yedoma permafrost as a neglected nitrous oxide source”

Maija E. Marushchak et al.

September 01, 2021

We thank the editor and all three reviewers for acknowledging the importance of this study and for thoughtful reviews. The comments helped us to improve the manuscript markedly. We hope that you will find our revisions are sufficient and satisfactory and may recommend publication after these changes have been made. Below we respond to the reviewer comments with blue italic font. All line numbers refer to the clean version of the revised manuscript without tracked changes.

Reviewer #1 (Remarks to the Author):

Permafrost affected soils are well recognized sources of the greenhouse gas methane and highly vulnerable to climate change. Nitrous oxide emissions were traditionally considered neglectable. Such a view is changing during the last decade, and Marushchak et al. nicely demonstrate that certain ‘hot spots’ in the permafrost zone are important factors to be considered when it comes to modelling of climate change. The present study is indeed timely, well conceived, highly interesting and nicely demonstrates in situ nitrous oxide emissions from vegetated, aged Yedoma permafrost. Even more important is the demonstrated potential of the sampled soils to produce large amounts of nitrous oxide when the conditions become more favourable for the processes involved, i.e., nitrification and denitrification.

Thank you for these positive comments and acknowledging the timeliness of the study!

The study addresses parameters affecting in situ nitrous oxide emission along with in depth microbiological analyses of associated organisms. During the presentation of the data, information on key microbes associated with nitrous oxide emissions was only touched upon. I would encourage the authors to provide more information on the relevant functional gene diversity obtained, potential key taxa and deposit sequencing data (provide accession numbers), if not already done. To provide figures and tables on the microbiological data in the supplementary materials is obviously an option.

Thank you for these opinions concerning the microbial data. We have now added a new Supplementary Fig. S9, where we have plotted relative abundances for those genes and gene ratios that were not shown in the Fig. 4. In the main text, we have added text where we elaborate the patterns in these additional genes (lines 255-267). We hope this decision pleases the reviewers and the editor.

The scope of this article remains, however, a biogeochemical one, which allows us to present a concise story without the need to compromise on the depth of discussion of the data currently shown. This made us to leave the discussion about the molecular results on the gene level without going to the detailed species level. Since the probe capture tool targeted 9 different N cycling genes, thorough analyses including key taxa, diversity and relation to environmental variable for all captured genes would have required more space than was possible in the frame of this article.

We do think that thorough analysis of the microbial data will be of a great value and we are planning to publish a follow-up paper in a microbiology journal, accompanied by the full sequencing data published in an open database.

Taken together, this well conceived study with attention to detail has a high degree of novelty and represents a major step forward.

We thank for this encouraging summary!

Specific comments:

L1 and throughout the manuscript. Shouldn't "Yedoma" and other proper nouns start with a capital?

We have seen Yedoma written both ways, capitalized and non-capitalized. After checking once more in the literature, the earlier convention seems to be more common, as the reviewer was suggesting. We have now changed this and used the capitalized form throughout the manuscript.

L34 Please avoid first time claims. The study provides obviously highly novel data; otherwise it shouldn't be published ...

We have removed this wording from the abstract.

L92/L118 Negative nitrous oxide emission rates indicate that such soils might represent a sink for nitrous oxide under special conditions. I would encourage to evaluate this finding (development stage, vegetation, soil parameters, microbiome, etc.) and add a short statement on this topic.

Thank you for pointing this out. Indeed, even small N₂O uptake rates when occurring over large areas may have the potential to compensate for high N₂O emissions occurring from limited area. Based on the current data, we wrote the following short summary, which is now included in the main text (lines 112-120):

"Of all measured N₂O fluxes, 23% were negative, with highest uptake rates < -25 μg N m⁻² day⁻¹ observed from bare Yedoma sites with high soil water content (WFPS = 67–100%; Fig. 2A). Uptake of atmospheric N₂O is commonly observed in wetland soils, where energetically more favorable electron acceptors, such as O₂ or NO₃⁻ are absent 6, 23. However, we occasionally measured small negative N₂O fluxes also from vegetated

Holocene cover with very dry topsoil (Fig. 2A; WFPS = 6–15%), supporting previous observations of N₂O uptake in dry oxic soils 24, 25. From all studied microsites, the median N₂O flux was negative only in vegetated Holocene cover in Kurungnakh (–4 μg N m^{–2} day^{–1}) and in freshly thawed Yedoma in Chersky (–9 μg N m^{–2} day^{–1}) (Supplementary Table S1).”

However, due to low number of negative fluxes, it is not possible to draw solid conclusions about commonness of N₂O uptake or its drivers. More data are obviously needed for proper understanding, and we are working with this question in follow-up projects

L97/98 repetition of “already”; please avoid.

We have removed the second occasion.

L128 What is the proportion of vegetated Yedoma relative to total Yedoma? Is there any information on how long such “hot spots” of nitrous oxide emissions remain stable?

These are very good questions and relevant for understanding the significance of the observed N₂O emissions from thawed Yedoma. We do not have quantitative information yet about the distribution of unvegetated vs. revegetated Yedoma but are currently working on upscaling for regional estimates in a follow-up project.

Similarly, we do not have any information about the persistence of the N₂O emissions with time. It is possible and even likely that the emissions cease with time with development of dense vegetation cover and decrease in net N mineralization after initial high rates, as we state this starting on line 307: “Although N₂O emissions may decrease with further slope stabilization as a result of continuous N losses and establishment of full vegetation cover...”. More research is needed to answer these questions and produce the first regional estimates of the N₂O emissions from Yedoma thaw slumps.

L137 Shouldn’t “Baydzherakhs” start with a capital letter?

This term is not very commonly used in the literature, but in the few occurrences it is usually uncapitalized. We would like to stick to this convention and keep it as is.

L213 Should this read: different microbial community structure related to N-cycling ?

Changed accordingly.

Furthermore, some more details on the microbial community structure (see comment above) and discussion in context of microbiological literature on nitrous oxide production associated permafrost microbiomes would fit here and after the next paragraph.

We agree that it is good to bring some more light on the other captured N cycling genes, and to explain how our key microbial results relate with literature. We have added following comparison with the previous findings of changes in N cycling genes associated with permafrost thaw (line 238-241):

“The sparse studies that have previously reported changes in N cycling genes associated with permafrost thaw are in line with our findings here: Significant increases in denitrification genes (norB, nirS, nosZ) have been observed following thaw^{30, 46}.”

Further, we have added the following gene results in the text (line 255->):

“In addition to the above-mentioned denitrification genes, also the relative abundance of the nrfA gene encoding nitrite reduction to ammonia was very low in vegetated Holocene cover (0.4%), intermediate in bare Yedoma (1.7–1.8%) and highest in revegetated yedoma (2.0–3.2%; Supplementary Figure S9). The nrfA gene is a key functional gene in dissimilatory NO₃⁻ reduction to ammonium (DNRA), and its increasing abundance with post-thaw age reflects the improved availability of NO₂⁻ from nitrification. The norB gene responsible for nitric oxide (NO) reduction to N₂O did not show similar gradual increase from bare to revegetated Yedoma sites as other nitrification and denitrification genes (Supplementary Fig. S9). This might be a result of a methodological bias: since norB has been less studied than nir and nosZ⁴⁷, the gene databases used for developing the probe capture tool are not including enough probe diversity for norB to cover arctic variants. Also, the nitric oxide reductase encoded by norB has a role in detoxification of NO, giving this enzyme a broader importance than just catalysing an intermediate step of the denitrification pathway⁴⁸.”

L216-227 What about the nor genes (same question to Fig. 4)? Nitrous oxide is the end product of Nor rather than Nir. Did the nor genes follow a similar trend like the nir genes?

We have added in the supplementary material the Supplementary Figure S9, which shows the relative abundances of the genes and gene ratios not shown in Fig. 4. This includes the norB gene. The relative abundance of the norB gene did not follow as clear increasing trend with post thaw age as we observed for the nir gene. It was low in vegetated Holocene cover (1.6%), but slightly higher in bare, recently thawed Yedoma sites (10.2–10.6%) than in earlier thawed and revegetated Yedoma sites (7.1–7.7%). However, the spatial pattern in norB/nosZ ratio was similar to that observed in (nirK+nirS)/nosZ ratio. We have summarized the patterns in norB gene in the text (lines 255-267; see above).

It is important to note that we likely have not captured the norB gene as efficiently as the nir genes, because the norB gene has been much less studied. Thus, the gene databases used to develop the probe capture tool are less representative for the norB gene than for the nir genes (e.g. Banerjee and Siciliano, 2012). This supports the use of nirS and nirK as functional gene markers for N₂O producing microbes. We have added this to the text (lines 260-267).

L240 Duplicate “the”.

Removed.

L458 Duplicate “S1”.

Removed.

L503 Fragmented sentence “We 500 μ l ...”.

Verb added.

L518 Please provide an explanation here, why only Kurungnakh soils were utilised for incubation experiments.

As we mention the first time on line 88, Kurungnakh was our primary site where we conducted the full set of measurements including N transformation rates and molecular studies, whereas in Duvanny Yar we only determined the field flux rates, soil gas concentrations and soil mineral N content. This is mainly due time limitations and challenging logistics related to the field work in Duvanny Yar and sample export. We have now added the following explanation to the first occasion, where we explain measurements conducted only at the primary study site: the section about Gross and net N transformation rates (lines 566-576):

“Nitrogen transformation rates were determined in the field laboratory from freshly sampled soils to imitate the processes occurring in the field during the flux measurements as realistically as possible. Due to time limitation and logistical challenges related to the field work in Duvanny Yar, we did not measure N transformation rates from Duvanny Yar soils, but only from soils sampled in our primary study site Kurungnakh.”

In the following sections about incubation experiments (lines 594-596) and molecular studies (lines 624-625), we again mention that Kurungnakh was our primary study site, with a reference to the N transformation section.

L533 Duplication of “constant”.

Removed.

L555 Shouldn't this read Table S5? Please check all references to data sets carefully.

Thank you for spotting this. We went carefully through all the references to figures and tables and made a few corrections.

L565 Which nir genes were included? nirK and nirS? Please clarify.

Both nirK and nirS were captured and were summed up for data presentation. We have added this info to a few occasions in the main text, the Methods section, and the captions of Fig. 4. and Supplementary Fig. S8. Further, we changed all occasions with nir/nosZ ratio to (nirK+nirS)/nosZ throughout the manuscript.

Figure 2 b. “100.0” etc. One digit might suffice here (“100” etc.). Please define ND in the legend.

We removed the digit from the y-scale, and defined ND in the legend of Figs. 2 and 3.

Reviewer #2 (Remarks to the Author):

Comments on the manuscript of Marushchak et al submitted to Nature Communications

The manuscript presents the results of a study on soil N cycling and N₂O emissions in two sites of Yedoma permafrost soils. They elegantly show, along a gradient of permafrost thawing, that nitrification and denitrification processes, which were absent in recently thawed permafrost, are slowly activated with soil exposure to surface microbes and plants. This reactivation is associated to large N₂O emissions, a power greenhouse gaz. Their results confirm a recently published study by Monteux et al. demonstrating the microbial limitations of soil N cycling in Yedoma permafrost using an inoculation approach. These last convergent news from the scientific community represent a major advance in our understanding on interactions between global warming, N cycling in arctic terrestrial ecosystems and feedbacks on climate. Therefore I fully support the publication of this study. Below some important comments that must be accounted before publication.

Thank you for these positive comments!

L34/35 the first time is incorrect since you confirm the findings of Monteux et al. Moreover, I think the message you carry out is sufficiently important to avoid such artifices. See below advices to how better position your studies regarding to the previous study.

Based on the comments from Reviewers 1 and 2, we have removed the claim about first findings.

L56/57 This sentence is tendentious and dangerous in term of message to the society (if natural ecosystems are an important source of GHG, why should we protect them and make effort to decrease anthropogenic GHG). Natural ecosystems and agricultural soils simply do not occupy same surface.

We agree that with the on-going climate crisis, it is important to pay attention to delivering the right messages about the importance different emission sources for GHG mitigation strategies. We reported the global N₂O emissions from soils as general background information and as a bridge between the general importance of atmospheric N₂O and our study about N₂O emissions from natural soils. However, we see now that not mentioning the importance of agricultural emissions may be misleading for those readers who do not know the global N₂O budget, since those emissions are responsible for the increase in atmospheric N₂O concentration during last decades and should be the primary target of mitigation strategies (Tian et al. 2020). We thank for pointing this out, and have revised the text as follows (lines 56-68):

“The current increase in atmospheric N₂O concentration is mainly driven by the growth of human-induced emissions, which comprise 43% of the global N₂O emissions of 17.0 Tg N yr⁻¹ and are dominated by N₂O release from fertilized agricultural soils¹³. Nitrous oxide emissions, although generally smaller per unit area, occur also from soils under natural

vegetation with a 33% contribution to the total global N₂O emission¹³. Tropical soils with high N turnover rates generally show the largest N₂O emissions among natural soils, while permafrost-affected soils in cold environments have been thought to be negligible N₂O sources. This view was challenged by a recent synthesis showing that N₂O emissions commonly occur from permafrost soils, with a global emission between 0.08 and 1.27 Tg N year⁻¹, meaning a 1–23% addition to the global N₂O emission from natural soils⁶.”

L94-98 Could you refer to the position on the slope (top -> down) when you describe your gradient. That will help the understanding for non-specialists.

It is indeed good to clarify this. We have edited the sentence and it reads now as follows (lines 99-104):

“The N₂O emissions showed an increasing trend along the measuring transect (Fig. 2A, Supplementary Table S1) from the densely vegetated Holocene cover deposits overlying intact permafrost on the top of the riverbank, through the actively eroding upper part of the Yedoma exposure, down to the already stabilized lower part of the slope revegetated by mosses and grasses.”

L179-190 I do not understand you invest of most your space to describe N mineralisation/immobilisation along the gradient instead of net nitrification (one line only for this process). Indeed, the change in nitrification along the gradient is higher than that of N mineralisation and this former variable is the most relevant regarding the message you convey (reactivation of nitrification and denitrification).

We agree that nitrification is very important for the occurrence of high N₂O fluxes from post-thaw Yedoma. However, we believe that it is not irrelevant to discuss mineralization which is the source of mineral nitrogen for microbial activities, including nitrification and denitrification. Although high N mineralization as such will not guarantee high N₂O production, substantial N₂O emissions will not occur if N mineralization rate is very low. Also, we think it is important to highlight the role of post-thaw N mineralization vs. direct mineral N release from thawing permafrost as a persistent source of mineral N (lines 185-187). We have read the text again with a good thought and think that is quite well balanced. We discuss nitrification throughout this and previous section (lines 144-156, 177-182, 188-193, 204-206), not only the final statement saying that: “The increasing trend with post-thaw age was even stronger for net nitrification than for net mineralization.”

L193-194 The use of terms “rhizosphere priming effect would be more adapted” in your case. The acceleration is not only caused by energy-C. This rhizosphere priming effect is the consequence of root activities including rhizodeposition, nutrient uptake, soil aggregates disturbance... You should cite the study of Henneron et al 2020 in New Phyt who present different rhizosphere priming and impacts on soil N fluxes induced by perennial grasses, which may explain the pattern observed.

We thank for advising us on this excellent reference and on correct use of terminology. We have now added the reference and revised the text, which now reads as follows:

“Despite the fact that plants can effectively compete for N species with soil microbes and suppress N losses by inhibition of nitrification and denitrification processes ³⁶, plants may also promote soil N cycling processes. Rhizosphere priming ³⁷⁻³⁹ is the term used for the summed effects of different mechanisms by which plants enhance SOM decomposition. These mechanisms include rhizospheric deposition of labile carbon compounds, which provide an easily available source of energy and C, and organic acids, which help to release protective organic-mineral associations, as well as the effect of roots on soil aggregation. Positive priming of N mineralization has been previously reported in high-latitude ecosystems ^{40, 41} and in nutrient-rich croplands grown with perennial grasses ⁴².”

L240-242 This is not correct, this is not the first evidence of the microbial limitation of N cycling in permafrost. A better positioning is needed here to highlight the novelty of your work: your findings support the conclusion of Monteux et al. on the microbial limitation of soil N cycling Yedoma permafrost obtained in an incubation study (in your case it's even not a demonstration, but a correlation). Moreover, you demonstrate in situ (for the first time, if you want to specify that) that this limitation can be levered in some years when thawed yedoma permafrost are exposed to surface microbes and plants. To my opinion this last finding is highly important for societal aspects.

We thank for this comments and have revised the text accordingly (lines 279-283):

“Taken together, our results represent the first in situ evidence for the microbial limitation of N cycle and N₂O emissions from thawing Yedoma permafrost due to low abundance of ammonia oxidizers, confirming the findings of a recent laboratory incubation study, which discovered this phenomenon ⁴⁹.”

L257-260 despite several reading I did not understand this sentence which are cut too many times by informations included in brackets.

Thank you for pointing this out, the sentence was complicated. We revised this part as follows, and hope it reads better now (lines 310-314):

“According to our remote sensing analysis using ArcticDEM and UAV data, the Yedoma cliff of Kurungnakh retreated in 2012–2019 as a result of permafrost thaw at a rate of 3.7 (2.5–5.7) m year⁻¹ (median with (25th–75th percentiles); Supplementary Fig. S3; see Methods section for details). Based on typical ice content of Yedoma deposits and the total N content of freshly thawed permafrost, we could estimate that this retrogressive thaw liberated at the thaw front as much as 1.7 kg of total N per m² per year, which was associated with a release of 39 g of mineral N per m² per year (see Methods section for details).”

L281 please correct in line with my previous comments.

For clarity, we split the sentence into two, and hope it reads better now (lines 337-341):

“Similar disturbed N-rich Yedoma with successional plant cover are widespread along thermokarst lake shores, coasts, slopes and valleys across the Yedoma region (Supplementary Fig. S11). Widespread occurrence of such landforms suggests that our findings are the first indication for substantial N₂O emissions over large areas in the Arctic.”

Reviewer #3 (Remarks to the Author):

Thawing yedoma permafrost as a neglected nitrous oxide source

Marushchak M.E., Kerttula J., Diáková K., Faguet A.4, Gil J.1,5, Grosse G.6,7, Knoblauch, C.8,9, Lashchinskiy N.4,10, Nykamb M.1, Martikainen P.J.1, 5 Morgenstern A.6, Siljanen H.M.P.1,11, van Delden L.1,6, Voigt C.1,12, Zimov N.13, 6 Zimov S.13, Biasi C.1

Comments to the authors

This paper, entitled ‘Thawing Yedoma permafrost as a neglected nitrous oxide source’ by Marushchak et al. addresses the complex changes in soil process occurrence and –rates in thaw gradients of Yedoma permafrost soil. N₂O is a greenhouse gas with ~300 times the global warming potential of CO₂, the emission of which has been wrongly assumed negligible in permafrost areas (Voigt et al 2020, Nature reviews Earth & Env.). This assumption, combined with the logistically challenging fieldwork, has led to scarce N₂O field data availability for the permafrost area. The field site selected here is representative for Yedoma soils, which contain large carbon (C) and nitrogen (N) stocks that have been immobilized (frozen) for thousands of years. With the biotic awakening of the arctic, these stocks of C and N now start to play their role in the dynamic part of global elemental cycling, and are as such likely a key-player in the permafrost-climate change feedback. The current study presents both measured N₂O emissions from relevant field sites as well as data essential to improve our understanding of the dynamic drivers behind N₂O emissions (mineralization, nitrification, denitrification, soil moisture, microbial community composition) in a changing climate. The authors also provide an estimate of the rate of overall N liberation from Yedoma thaw slumps, a first of its kind as far as I am aware. Moreover, the authors have fitted their experimental design to be able to test, in a field setting, new and exciting hypotheses thus far only based on laboratory incubations (nitrification in freshly thawed permafrost soils hampered by functional limitation of the microbial community, Monteux et al 2020, Nature geosci).

This study provides important field data on both the magnitude of N₂O emission (133–6286 µg N m⁻² day⁻¹) and, more general, N-liberation (1.7 kg m⁻²yr⁻¹) from thawing Yedoma permafrost soils, as well as of the (short, < 10 years) time it takes for ‘functionally limited’/‘virgin’ freshly thawed permafrost soils to become colonized by external microbes. These data are a strong indication that potential estimates of N₂O emission based on laboratory incubations using ‘freshly thawed’ permafrost soils without inoculation with external microbes might underestimate actual emissions in the field. The non-linear processes

shown here are currently not taken into account by Earth System Models and this study will give direction to ongoing discussions between field-researchers and EMS modellers.

I find this paper well-written, the data presented in the main text excellent and the supplementary material information all of interest. I did not find flaws in data-analysis, interpretation and conclusions. This study seems to be carefully performed by a brave team of researchers, who took the important and challenging work upon them to get these data out. The authors are not overstating the importance of their work, and they put their data in appropriate perspective.

Thank you, we are very happy to receive this positive feedback!

Minor comments:

I. 483 'bluk soil' -> bulk soil

Corrected.

I. 228 " to demonstrate that", please rephrase to (something like) " to test whether..."

Changed as suggested.

I. 242 In reference 41 and 42 functional limitation of freshly thawed permafrost soil is discussed, and nitrification, (part of) respiration, and methanogenesis are hypothesized as potentially missing functions. You specifically test for and discuss nitrification. But you also seem to have CH₄ and CO₂ data for the same thaw gradient on which you base your estimate of 'colonization time'(my words) by external microbes (i.e. regaining soil functioning). I understand that it might be beyond the scope of this paper, but since you have the results, would it be possible to discuss your CH₄ and CO₂ results shortly, either in the main text or in a short supplementary discussion? I.e. are your findings in line with what was previously observed/hypothesized and if not, what are your thoughts on why?

There would be definitely a lot more to discuss in the C flux data associated with our N₂O measurements, but as said by the reviewer, this is beyond the scope of this paper. However, encouraged by this comment, we decided to add short comments about CO₂ and CH₄ fluxes into the main text in appropriate places.

About CO₂ on lines 223-227: "However, it is difficult to separate the plant effects on N₂O emissions from the effects of moisture changes. Similarly, the increase in ecosystem respiration rates with slope stabilization and revegetation was observed in both exposures (Supplementary Fig. S6, Supplementary Table S1), reflecting the joint effect of drying and increased plant C input, which cannot be distinguished from each other at this stage."

About CH₄ on lines 284-297: "Similarly to ammonia oxidisers in the present study and in the earlier laboratory study 49, it has been shown previously that also methanogens represent a bottle-neck in yedoma permafrost biogeochemistry 50: a small part of the microbial community carrying out an important function associated to permafrost-climate feedbacks."

On the studied yedoma exposures, the low CH₄ emissions from bare freshly thawed yedoma (< 0.5 mg C m⁻² d⁻¹) despite the high water saturation suggest that also there methane production was limited by lack of methanogenic archaea (Supplementary Fig. S6, Supplementary Table S1). Most of the studied soils showed minor CH₄ emission or consumption rates (Supplementary Table S1), and even the highest median CH₄ emission 1.95 mg C m⁻² d⁻¹, observed in revegetated yedoma with mosses in Kurungnakh, was modest compared to the emissions from polygonal wetlands (2–35 mg C m⁻² d⁻¹) 51 and small ponds (4–35 mg C m⁻² d⁻¹) 52 in the same region. Lack of recovery of methanogenic function with post-thaw age is easy to explain with drying associated with slope stabilization at such Yedoma exposures 19, which creates unfavorable conditions for anaerobic methanogens.”

General:

- Perhaps address shortly in the supplementary methods attention points related to selecting a time and a place for N₂O measurements (hot spots, hot moments) and potential implications.

It is very true that high spatial and temporal variability cause serious challenges regarding representativeness of data collected with manual chambers during short campaigns as we did here. However, the robust patterns we observed in the spatial distribution of N₂O fluxes in Kurungnakh (where two measurement rounds were conducted) and our ability to explain these spatial patterns so well with microbial and environmental data show that our study design was successful. Here, we benefit from our previous experience with collecting N₂O data from permafrost-affected soils. We have added to the Supplementary methods section the following short text about our thoughts on dealing with spatial and temporal heterogeneity in soil N₂O flux studies.

“Although N₂O emissions are known to commonly occur as high pulses or hot moments often outside the peak summer period 9, 10, our own studies on bare peat surfaces have shown that temporal pattern of the emissions in these hot spots follows closely the temperature variations, peaking at high summer 11. This (besides obvious logistical reasons) guided our selection of timing for the field campaign since we wanted to catch the peak emissions and thus maximize the emission range to reveal the drivers of spatial variability. The two measurement rounds in Kurungnakh confirm that the spatial patterns observed in N₂O emissions were robust with respect to timing: the correlation between the two measurement occasions was very high (Pearson correlation with log-transformed data, $R = 0.92$, $p < 0.001$; data not shown). Another challenge related to soil N₂O measurements is the large spatial variability 10. Plot selection for manual chamber studies requires usually compromising between the number of studied surface types and the replicate number; here, we used the replication of five, which has proved sufficient in our previous studies 12.”

- Data stored at zenodo (<https://zenodo.org,doi:10.5281/zenodo.4587235>) are not actually available for reviewing – it is needed to apply for access from the authors. This is perhaps too time-consuming and, why not, awkward, to be useful for reviewing purposes?

We apologize for selecting the wrong access category for our data in Zenodo, it was not intended that application for access is needed. The doi has been now updated with link to the openly available data sets.

Other changes:

- *We accidentally missed one of the coauthors, J.G. Ronkainen, from the author list of the previous version; for this version we have added him.*
- *Supplementary Fig. S11 (Previous Supplementary Fig. S10) has been complemented with a map and additional aerial photos showing sites with surfaces with high potential for N₂O release.*
- *Section ‘N losses from thawing Yedoma and implications of permafrost N release’ (lines 314-317 & 329-333): We noticed an error in our calculations of annual N₂O compared to the total N release associated with retrogressive thaw at the thaw front: the N₂O emission factor of 3% of the total soil N mobilization was too high, and we apologize for not noticing it before the initial submission. We have now recalculated the emission factor using this time the direct mineral N release from permafrost as reference data for N₂O emissions instead of bulk N. The corrected emission factor is much lower than we had earlier calculated (0.14%), but we still think that >1% release of liberated mineral N within ten years, if conditions are favorable) is rather high for natural soils (IPCC emission factor for agricultural soils is 1%) and worth mentioning in the main text. The corrected text reads as follows:

“Based on typical ice content of Yedoma deposits and the total N content of freshly thawed permafrost, we could estimate that this retrogressive thaw liberated at the thaw front as much as 1.7 kg of total N per m² per year, which was associated with a release of 39 g of mineral N per m² per year (see Methods section for details).”...

... “Based on the emissions from disturbed Yedoma revegetated with grasses on Kurungnakh (median multiplied with a snowfree-season-length of 100 days), we estimated that under these optimal conditions thawed Yedoma will release 54.8 mg N m⁻² just in one year. This corresponds to 0.14% of the mineral N originally liberated at the permafrost thaw front from a similar area in a year (see above). This is 7 times lower than the IPCC N₂O emission factor for N fertilization in managed mineral soils (1%)⁵⁷, but still high considering that it occurs in a pristine northern soil, which are generally N limited²⁸.”*
- *The Methods (lines 686-692) and Supplementary Methods sections have been changed accordingly.*

- *Other minor changes (typos etc.) can be seen in the manuscript file with tracked changes.*

REVIEWERS' COMMENTS

Reviewer #1 (Remarks to the Author):

The authors adopted most of my suggestions and made a well written manuscript even better. The only step that needs to be taken by the authors before publication is submission of the sequence data to public databases (Genbank or EMBL, etc.) to make the data available.

Reviewer #2 (Remarks to the Author):

I confirm here the novelty and the importance of this study regarding to the central question of positive feedbacks between global warming and permafrosts. Overall, the modifications brought by the authors responded to my comments. Just some minor comments that do not need further checks from my side:

- I am wondering whether you could insist a bit more on the idea that thawed Yedoma become a large source of mineral N, in addition to of course your message on N₂O emissions. I have the feeling that the large amount of mineral N can have a huge impact surrounding on aquatic environments (eutrophication). This might some additions in abstract, title?

-L345 in the manuscript version in correction mode, you should specify we are talking about N₂O emissions and not mineral N release. This is not obvious.

-I suggested to you the reference Henneron et al because they show strong influence of plant nutrient economy (conservative<->acquisitive) on soil N mineralisation. I was wondering whether the change in vegetation within the transects and among studied sites may explain the shift in soil N mineralisation. But it is maybe too speculative at this step, I just wanted to share this idea with you.

Reviewer #3 (Remarks to the Author):

With pleasure I read the rebuttal letter by M E Marushchak et al. I still support the publication of this manuscript in Nature communications. My comments were all addressed, and I find that the manuscript benefitted from all reviewers' comments.

Response to the second round of reviewer comments for the manuscript NCOMMS-21-09918-T, titled “Thawing Yedoma permafrost is a neglected nitrous oxide source”

Maija E. Marushchak et al.

October 26, 2021

We thank the three reviewers and the editor for approving the revisions made during the first revision round and supporting publication of this study. Below we respond to the reviewer comments and list other changes made to the manuscript with blue italic font. All line numbers refer to the revised manuscript version with tracked changes.

Reviewer #1 (Remarks to the Author):

The authors adopted most of my suggestions and made a well written manuscript even better. The only step that needs to be taken by the authors before publication is submission of the sequence data to public databases (Genbank or EMBL, etc.) to make the data available.

We are happy that our revisions and comments were satisfactory. We have now submitted the metagenomics data to a public database as suggested, and added the following sentence in the Data availability statement: “The metagenomic data are deposited to the SRA database under the BioProject link PRJNA771879.”

Reviewer #2 (Remarks to the Author):

I confirm here the novelty and the importance of this study regarding to the central question of positive feedbacks between global warming and permafrosts. Overall, the modifications brought by the authors responded to my comments. Just some minor comments that do not need further checks from my side:

Thank you!

- I am wondering whether you could insist a bit more on the idea that thawed Yedoma become a large source of mineral N, in addition to of course your message on N₂O emissions. I have the feeling that the large amount of mineral N can have a huge impact surrounding on aquatic environments (eutrophication). This might some additions in abstract, title?

Although this is a very valid point, we think it is beyond the scope of this article to discuss the effects for surrounding aquatic ecosystems. We hope to see such studies future that tackle this important question.

-L345 in the manuscript version in correction mode, you should specify we are talking about N₂O emissions and not mineral N release. This is not obvious.

Very true. We have now clarified this as follows:

“...we estimated that under these optimal conditions thawed Yedoma will lose 54.8 mg N m⁻² of N₂O to the atmosphere just in one year”

-I suggested to you the reference Henneron et al because they show strong influence of plant nutrient economy (conservative<->acquisitive) on soil N mineralisation. I was wondering whether the change in vegetation within the transects and among studied sites may explain the shift in soil N mineralisation. But it is maybe too speculative at this step, I just wanted to share this idea with you.

This is an exciting idea; we thank for sharing it with us. However, we wanted to be careful not to go beyond the evidence provided by the current data when discussing plant-microbe interactions. Again, definitely something worth investigating in future.

Reviewer #3 (Remarks to the Author):

With pleasure I read the rebuttal letter by M E Marushchak et al. I still support the publication of this manuscript in Nature communications. My comments were all addressed, and I find that the manuscript benefitted from all reviewers' comments.

Thank you!

Other corrections:

We modified the structure and technical details of the manuscript to match with the journal guidelines and the author checklist. Additionally, we did the following corrections:

line 39: The median N₂O emission from permafrost affected soils is 38 μg N m⁻² day⁻¹, so the emissions from grassy Yedoma sites are 1-2 orders of magnitude higher than this, not 2-3 orders of magnitude, as we had previously written.

lines 305-306: The section title has been slightly modified to better match with the content.